# Repetitive Fragile Sites: Centromere Satellite DNA as a Source of Genome Instability in Human Diseases

**DOI:** 10.3390/genes9120615

**Published:** 2018-12-07

**Authors:** Elizabeth M. Black, Simona Giunta

**Affiliations:** Laboratory of Chromosome and Cell Biology, The Rockefeller University, 1230 York Avenue, New York, NY 10065, USA; elizabethblack432@gmail.com

**Keywords:** centromere, alpha-satellite, genome stability, repetitive DNA, human genome, recombination, fragile sites, genome instability, replication, cancer, aging

## Abstract

Maintenance of an intact genome is essential for cellular and organismal homeostasis. The centromere is a specialized chromosomal locus required for faithful genome inheritance at each round of cell division. Human centromeres are composed of large tandem arrays of repetitive alpha-satellite DNA, which are often sites of aberrant rearrangements that may lead to chromosome fusions and genetic abnormalities. While the centromere has an essential role in chromosome segregation during mitosis, the long and repetitive nature of the highly identical repeats has greatly hindered in-depth genetic studies, and complete annotation of all human centromeres is still lacking. Here, we review our current understanding of human centromere genetics and epigenetics as well as recent investigations into the role of centromere DNA in disease, with a special focus on cancer, aging, and human immunodeficiency–centromeric instability–facial anomalies (ICF) syndrome. We also highlight the causes and consequences of genomic instability at these large repetitive arrays and describe the possible sources of centromere fragility. The novel connection between alpha-satellite DNA instability and human pathological conditions emphasizes the importance of obtaining a truly complete human genome assembly and accelerating our understanding of centromere repeats’ role in physiology and beyond.

## 1. Current Overview of Human Centromeric DNA

Cells are equipped with sophisticated molecular networks that ensure faithful transmission of genetic information at each round of cell division. Chromosome segregation relies on the centromere, which connects chromosomes to the spindle microtubules that are responsible for separating sister chromatids. Centromere function is conserved across the eukaryotic kingdoms, but their sequences are not. One unifying feature of centromeres throughout evolution is the prevalence of repetitive DNA sequences, with some notable exceptions, such as the 125 base pair (bp) point centromere found in budding yeast (reviewed in [1]). However, the presence of neocentromeres in nonrepetitive genomic loci strongly supports the idea that this repetitive feature may not be essential for maintaining centromere function (reviewed in [2]), which raises multiple questions: (1) why repetitive centromeric DNA is prevalent; (2) what potential advantages may arise from forming the centromere on DNA repeats; and (3) whether there is an active mechanism to preserve repetitive DNA at the centromere. Highly homogenous repeats have challenged current sequencing and assembly technologies and, thus far, precluded the complete linear assembly and annotation of all centromeres. Indeed, The Human Genome Project, marked as completed in 2003 [3], actually omitted over 10% of the human genome, including large portions of the centromere and other repetitive elements. A linear assembly for the Y centromere has been achieved recently [4], and reference models for other centromeres have been established ([5,6] Miga et al. unpublished data), which may pave the way for future analysis of human centromere genetics. Nonetheless, with only partial annotation available in the reference genomes for most centromeres, in-depth functional studies of these repetitive elements have been greatly hindered.

Human centromeres are defined genetically by the presence of head-to-tail tandem repeats composed of ~171-bp monomers of alpha-satellite DNA, which are further arranged into higher order repeats (HORs). Individual monomers of alpha-satellite DNA may be 50–70% similar, but HORs share 95–98% similarity [7,8]. Genetic variation within the HORs may have functional significance, epitomized in examples of epialleles. Epialleles are distinct HOR variants present on the same chromosome, both of which are capable of producing a fully functional centromere [9]. Importantly, in individuals heterozygous for the two epialleles, functional centromeres preferentially form on the array with less genetic variation [9], suggesting that sequence homogeneity promotes centromere assembly. The core centromere, made of highly homogenous HORs, is the functional site to enable chromosome division and spans from 0.5 to 5 Mb in size depending on the chromosome. The size of a particular array may vary by over an order of magnitude between individuals [10,11,12]. This variation, surprisingly, does not seem to have an effect on mitotic centromere function and faithful chromosome segregation. However, more in-depth studies are required to fully understand the effects of copy number on the function and stability of these repetitive regions, especially in light of recent data showing that repeats expansion confers centromere strength underlying meiotic drive [13], reviewed in [14]. While changes in repeat number at the rDNA locus, for example, have been extensively studied [15], the full consequences of variation in centromeric array size remain unclear.

In the regions flanking the centromere, the highly homogeneous HOR arrangement of alpha-satellite DNA becomes progressively more heterogeneous and unstructured. These regions are called pericentromeres and their organization and epigenetic makeup are distinct from centromeres. Within the pericentromere, alpha-satellite DNA is predominantly monomeric (no longer arranged into HORs) and interspersed with other sequences and repetitive elements, including the LINE and SINE retrotransposons (reviewed in [16]). Additionally, other types of satellite repeats, such as the 5-bp satellite II and satellite III repeats, are abundant in the pericentromere of certain chromosomes [17], reviewed in [18]. In contrast, alpha-satellite DNA is present on every centromere (reviewed in [19]).

Primate and mouse centromeric DNA contain the CENP-B box, a 17-bp motif that is bound by the centromere protein CENP-B. In humans, the CENP-B box is present on every chromosome except for the Y chromosome [20]. It is paradoxically necessary for formation of de novo centromeres on artificial chromosomes, but it is not essential for the formation of neocentromeres, and mice lacking CENP-B are viable and fertile [21,22].

## 2. Epigenetic Specification and Inheritance of Centromeres

The human centromere is epigenetically specified, maintained, and inherited transgenerationally by the presence of the centromere-specific histone H3 variant CENP-A (reviewed in [23]). CENP-A is deposited in interspersed domains among all functional centromeres [24,25]. The alpha-satellite HORs containing CENP-A represent the active array where the kinetochore is assembled (reviewed in [26]). A testament to the true epigenetic nature of the centromere, neocentromeres and centromeres on artificial chromosomes are also specified by CENP-A on DNA sequences unrelated to the endogenous centromeric locus [27]. CENP-A appears to preferentially bind about 10 bp upstream of the CENP-B box, but whether this is sequence specific or mediated by recruitment to CENP-B remains unclear [13].

The centromere has a unique epigenetic profile that is characterized by the presence of CENP-A and by a bivalent chromatin state that partially mimics active regions of the genome. Euchromatic post-translation modifications (PTMs) such as dimethylation of lysine 4 and 36 are found on the H3 nucleosome proximal to the CENP-A containing nucleosomes, yet there is a general lack of H3 and H4 acetylation typical of active chromatin [28,29,30]. Additional centromeric PTMs include monomethylation of histone H4 lysine 20 directly on CENP-A nucleosomes [31] and H3 lysine 9 and 27 dimethylation on H3-containing nucleosomes [32]. Hypomethylation of centromeric DNA has also been reported [33,34].

The pericentromere, in contrast, is a site of constitutive heterochromatin. Constitutive heterochromatin describes the highly compacted and transcriptionally repressed chromatin conformation found in repetitive and gene-poor genomic regions, such as telomeres (reviewed in [35]). The pericentromere is characterized by heavy DNA methylation and trimethylation of histone H3 lysine 9 (H3K9me3) (reviewed in [36]). Heterochromatin is important for suppressing centromeric recombination and modulating transcription [37], reviewed in [36], and thus may impact the overall integrity of these regions (see Section 4.4: “Breaking the Silence: Active Transcription of Centromere Alpha-Satellite Challenges Repeats Stability”).

Like its canonical counterpart H3, CENP-A undergoes different PTMs, such as phosphorylation, ubiquitination, and methylation [38,39,40]. Some of these modifications have been hypothesized to play a role in cell cycle coordination, for example, by orchestrating chromatin condensation in preparation for mitosis, influencing timing and initiation of replication, and modulating interactions with CENP-C, which effects changes to chromatin accessibility [40,41]. Specific PTMs have been implicated in the tight cell cycle regulation of CENP-A deposition, propagation, and inheritance (reviewed in [42]).

The activity, expression, and function of CENP-A vary significantly throughout the cell cycle. CENP-A loading onto centromeric chromatin occurs in a cell-cycle-dependent manner during late telophase/early G1 [43] in a highly regulated multistep process (reviewed in [44]) involving the CENP-A-specialized histone chaperone HJURP [45]. During S-phase, CENP-A is diluted equally in the daughter strands [43,46,47] and interspersed with H3.1- and H3.3-containing nucleosomes [47]. Behind the replication fork, CENP-A is precisely reassembled onto the same alpha-satellite repeats [48] through a recently identified mechanism that requires local tethering of CENP-A to the replication fork by HJURP and the MCM2 replicative helicase [49]. The incorporation of newly synthesized CENP-A into centromeric chromatin during telophase/early G1 correlates with the time at which RNA polymerase II (RNA Pol II) actively transcribes centromeric DNA [50,51]. This could suggest a coordinated role for centromeric transcription and CENP-A deposition [51].

A two-step mechanism can explain how CENP-A can identify, maintain, and propagate centromeres. The CENP-A-targeting domain (CATD) within the histone fold allows for CENP-A loading and subsequent centromere positioning. In turn, the N- and C-terminal tails serve as docking sites for the recruitment of CENP-C, stabilization of CENP-B, and assembly of the inner kinetochore platform [52], reviewed in [53].

In addition to centromere specification, we have recently shown that CENP-A is involved in maintaining alpha-satellite DNA integrity in cycling human cells [54]. Using the Cen-CO-FISH method [55], which can detect aberrant DNA rearrangements such as sister chromatid exchange at the centromere (C-SCE), CENP-A and associated proteins of the CCAN network CENP-C and CENP-T/W were found to suppress centromeric recombination [54]. This implies that these factors, which are constitutively associated with centromere loci throughout the cell cycle, may be part of a network that protects DNA repeats and promotes their stability.

## 3. Centromere Stability in Human Health and Disease

Physical breaks occur more frequently at the centromere and pericentromeric regions than at other regions of the genome ([56]; S.G., unpublished data). This preferential breakage is predicted to be due in part to the highly repetitive nature of the underlying DNA. Although determining a cause-and-effect relationship between human pathology and centromere alterations has been elusive, centromeric fragility has been strongly implicated in many aspects of health and disease.

### 3.1. Human Immunodeficiency–Centromeric Instability–Facial Anomalies (ICF) Syndrome

ICF syndrome is a rare genetic disorder caused by mutations in one of four identified genes: DNMT3b [57], HELLS [58], CDCA7 [58], and ZBTB24 [59]. Most cases of ICF syndrome (55%) are caused by mutation in DNMT3b, the gene coding for the human de novo DNA methyltransferase [60]. This disease causes severe immunodeficiency, increased susceptibility to infection, abnormal facial features, and cognitive disabilities [61,62]. ICF syndrome is often diagnosed by the presence of stretched and fragile juxtacentromeric heterochromatin on chromosomes 1 and 16 in activated lymphocytes. Perhaps as a result of this phenotype, these chromosomes are more susceptible to breakage, missegregation resulting in aneuploidy, and micronuclei formation [62]. Mutation in CDCA7, HELLS, and ZBTB24 also results in DNA methylation defects at alpha-satellites and satellite II DNA that is enriched on juxtacentromeric heterochromatin on chromosomes 1 and 16 [58,63]. HELLS and CDCA7 work together in a complex known as CHIRRC (CDCA7-HELLS ICF-Related nucleosome Remodeling Complex) to catalyze nucleosome remodeling, which could modulate the accessibility of DNA for methylation [64]. DNA methylation profiles in ICF patients with mutations in any of these three genes are different from that in ICF patients with DNMT3b mutation, indicating that these proteins may not all work in the same pathway [65]. All observed ICF patients, however, have hypomethylation of the juxtacentromeric satellite II repeats, leading to the hypothesis that the chromosome fragility and disease symptoms are directly linked to DNA hypomethylation.

The link between DNA methylation defects and centromere DNA stability is currently underexplored. Notably, DNA methylation has been correlated with suppression of centromeric recombination [37], and observed fragility at juxtacentromeric heterochromatin in activated lymphocytes from ICF patients suggests that DNA methylation could directly or indirectly regulate the size of the centromeric array or its structural integrity [61]. The cytogenetic phenotype of this disease indicates that epigenetic factors such as DNA methylation and chromatin remodeling may play important roles in centromere structure and function. However, it is worth noting that fragile juxtacentromeric heterochromatin is not observed in fibroblasts from ICF patients, and no apparent centromere instability is seen on chromosomes that do not contain the large satellite-II-rich juxtacentromeric heterochromatin, even in activated lymphocytes. Although our recent understanding of ICF syndrome has significantly increased, many outstanding questions remain, including the role of DNA methylation in the structural integrity of juxtacentromeric heterochromatin and the way in which the cellular phenotypes of ICF syndrome translate to patient symptoms, such as mental retardation, immunodeficiency, and facial anomalies.

### 3.2. Aging: Antagonistic Pleiotropy Applied to Centromeres

As cells age, the burden of genomic instability and ensuing DNA mutations may lead to deregulation of cell division and, ultimately, aging-associated diseases such as cancer. Thus, there are elaborate mechanisms in place to regulate cell division, especially in aging cells. The role of telomeres, the repetitive sequence at the end of linear chromosomes, has garnered much attention for its relation to aging. Telomere repeats serve as an internal clock for cycling cells because each round of replication results in the loss of telomeric DNA in the absence of active telomerase (reviewed in [66]). Eventually, this loss over cellular generations culminates in telomere crisis and a permanent state of cell cycle arrest called replicative senescence (reviewed in [67]). Interestingly, centromere deterioration has also been reported with advancing age [68]. Centromere instability has been associated with senescence arrest [69], but the connection with aging at the cellular or organismal level remains unclear. We recently reported enhanced rearrangements and sister chromatid exchange at centromere alpha-satellite DNA in cells approaching replicative senescence [54], inviting new questions about whether checkpoints may be in place that detect centromere dysfunction as cells age.

One possible explanation for the observed correlation between centromeric instability and senescence is the reduction or mislocalization of CENP-A away from the centromere, which has been reported in aging and cellular senescence [69,70,71]. Based on these data, we hypothesize that an age-related decrease in CENP-A may serve to trigger a senescence-like state of irreversible cell cycle arrest [54]. It has been suggested that this decrease or mislocalization of CENP-A may be mediated by changes in the transcription of the centromeric repeats [54]. While senescent cells display typical large foci of facultative heterochromatin, constitutive pericentromeric heterochromatin is instead decondensed [72], which could in turn hyperactivate transcription of these regions [73]. Indeed, hyperactive centromeric transcription can result in CENP-A mislocalization and mitotic arrest [69]. This may represent an important safety mechanism to prevent additional cell division in the presence of centromere instability. Future studies will be required to establish how compromising CENP-A and/or alpha-satellite DNA repeats trigger senescence and whether they can be recovered upon transformation or during the generation of induced pluripotent cells.

### 3.3. Cancer: The Multifaceted Role of Centromeres in Tumorigenesis

Karyotypic abnormality is a key feature of many cancer cells, including increased rates of aneuploidy, rearrangements, deletions, fragmentations, and duplications [74]. The severity of chromosome instability is closely correlated to tumor severity and poor prognosis (reviewed in [75]). Human centromeres are often sites of aberrant rearrangements in many tumors, causing chromosome fusions and genetic abnormalities [76,77] (Figure 1). Chromosome instability (CIN), frequently observed in human tumorigenesis, has been attributed to centromeric fission [78]. In agreement with studies that have shown connections between centromeric instability and cancer, we have reported that various cancerous cell lines display an increased rate of centromeric recombination and other abnormalities compared to healthy cells in a manner unrelated to chromosome missegregation [54]. Consistently, the enhanced centromere rearrangements reported are not limited to CIN cancer cell lines but are also found in karyotypically stable HCT116 cells. While HCT116 cells accurately segregate chromosomes [79], they are defective in intra-S-checkpoint activation during replication and in DNA damage repair [80]. Although cellular stress and abortive repair events during DNA replication are likely to cause centromere instability, additional distinct mechanisms may exist and/or converge to cause centromere instability in CIN and non-CIN cancer cell lines. While the frequency of centromere rearrangements in cancer is well appreciated, the mechanism of their malignancy is poorly understood. One possible explanation may lie in the perturbation of the epigenetic landscape of specific genomic regions, triggering their subsequent transcriptional dysregulation [81]. Rearrangements within the pericentromeric DNA seem to disrupt not only the local heterochromatin but also contribute to long-range changes to gene expression [81], suggesting that centromere DNA instability may similarly induce changes in the chromatin state of the region and result in aberrant rates of transcription contributing to malignant transformation.

Dysregulated transcription has been implicated in both centromeric instability and cancer, possibly as a result of compromised epigenetic silencing (reviewed in [82]). In some cancerous cells, noncoding RNA (ncRNA) transcribed from the satellite-II-rich pericentromeric regions may lead to increased formation of mutagenic RNA–DNA hybrids [83]. When unresolved, these hybrids can contribute to the expansion and instability of pericentromeric satellite repeats [83]. Given that these transcripts appear to only be produced in cancerous cells growing under nonadherent conditions, this may be exploited as a potential therapeutic treatment for centromeric or pericentromeric instability in cancer.

Another key link between the centromere and human cancers is the widespread overexpression of CENP-A across a variety of tumors [84,85,86,87]. Increase in CENP-A levels generally correlates with poor prognosis and tumor aggressiveness [88,89,90]. Several studies have detailed how CENP-A overexpression leads to its erroneous incorporation within ectopic interstitial loci outside of the alpha-satellite [48,91,92,93,94]. Interestingly, DNA replication has been recently identified to act as an error-correction mechanism that removes ectopically loaded CENP-A outside of the centromeres while retaining centromere-bound CENP-A [48]. Whether and how ectopic deposition of CENP-A results in hyperproliferation and tumor progression is unclear. A recent study, which shows that ectopic incorporation of CENP-A in the proximity of the *MYC* locus leads to hyperactive expression of this gene [93], may suggest one possible explanation for this connection. CENP-A expression is now routinely part of the breast cancer biomarker assessment for chemotherapy [95], which underscores the importance of further deciphering the connection between CENP-A and cancer. The correlation between CENP-A overexpression and cancer severity is well appreciated, yet it is unclear whether this overexpression leads to centromere instability. It is possible that, albeit the overall levels are increased, chromatin incorporation of CENP-A at centromeres is unchanged or decreased [S.G., unpublished observation]. This would reconcile the increase in centromere repeats instability seen across multiple cancer cells [54].

Centromeric instability can trigger carcinogenesis via different, well-established routes: (1) Compromised centromeric integrity may lead to an increased rate of DNA rearrangements, mutations, and overall genomic instability, which may subsequently lead to an increased likelihood of developing cancer. (2) Transformation to a cancerous cell may initiate centromeric instability by turning off certain checkpoints, resulting in increased rates of duplication, translocation, or breakage. (3) Centromere recombination and DNA rearrangements may impact the formation of a functional kinetochore, providing a direct mechanism by which centromere repeats instability drives chromosome missegregation, aneuploidy, and, in turn, tumor formation (Figure 1).

## 4. Sources of Instability within the Centromere DNA Repeats

### 4.1. Recombination and Repair at Centromeres: Errors in Copying and Mending Highly Repetitive DNA

“Why are centromeres so cold?”, asked Andy Choo in his review of centromeres [96]. He was referring to centromere DNA as being “cold” to recombination. While maternal and paternal chromosomes suffer multiple DNA double-stranded breaks (DSBs) to induce recombination and exchange of genetic information by crossing over during meiosis, centromere loci are refractory to this process. This led to the assumption that centromeres do not undergo recombination and that the repetitive arrays are maintained as stable. However, this clashed with the notion that centromeres’ very origin stems from recombination to create the repetitive array, where multiple short- and long-range recombination events may be responsible for the generation and reiteration of blocks of highly homogenized alpha-satellite DNA throughout the centromere [97,98]. Furthermore, in addition to chromosome-specific alpha-satellites, certain centromeric sequences are shared by all chromosomes, evidence that formation of these arrays is dominated by interchromosomal exchanges [8,98,99,100]. This invites new questions about the stability of centromere DNA outside of meiosis. Indeed, our recent analysis has shown that centromeres can undergo recombination during a single round of cell division in primary human cells [54]. Depletion of CENP-A and other CCAN proteins exacerbates centromere rearrangements [54], indicating that there may be active mechanisms to suppress centromeric recombination and these may, at least in part, involve core centromeric proteins.

Centromere alpha-satellite DNA is estimated to represent between 3% and 10% of the human genome [101], reviewed in [19]. During each round of replication, unperturbed cells suffer over 40 DNA DSBs [102], of which at least half are repaired by homologous recombination (HR) in S-phase and G2, when the sister chromatid is available as a template for repair (reviewed in [103]). Thus, it should not be surprising that centromeres stochastically suffer DNA damage, including DSBs, which may be repaired by HR in cycling cells. Due to the highly repetitive nature of the centromere, recombination here may be particularly perilous, as the HR pathway may easily identify nearly identical but nonhomologous sequences as the template for repair, leading to different forms of genomic instability (as outlined in Figure 1) that may directly impact chromosome integrity [104]. Homologous recombination between centromere repeats could generate aberrant centromere length, unequal or erroneous exchanges, and, ultimately, lead to alterations in the array, genome, and chromosome instability (Figure 1).

### 4.2. Secondary Structures: Physical Hurdles and Barriers to DNA Repeats Stability

Within the cell, DNA is under torsional strain from the topological arrangement of the duplex molecule into a right-hand turn, called B-form DNA. This coiling presents an advantage over non-twisted DNA, as it permits the dynamic opening of the helix with a lower energy barrier to allow for transcription, replication, and repair. However, non-B-DNA forms—namely, A and Z, as well as higher-order secondary and tertiary structures—are predicted to be rather prevalent in the human genome, particularly in highly repetitive regions [105], reviewed in [106]. This invites questions about their potential benefits and function, as well as whether active mechanisms exist to readily deal with these structures during DNA-based transactions. It has been proposed that the presence of noncanonical secondary structures is an identifying feature of centromeres [105]. This may offer a potential resolution to the CENP-B paradox, which refers to the discrepancy where the CENP-B box, the docking site for the CENP-B protein, is a necessary DNA component for the formation of de novo centromeres on artificial chromosomes but not on neocentromeres, and CENP-B itself is non-essential. The increased propensity for non-B-DNA forms around centromeres may present a recognizable substrate for CENP-B binding, an idea supported by the observation that the frequency of the CENP-B box motif is inversely proportional to predicted secondary structure formation [105].

Several non-B-form DNA structures have been identified within the alpha-satellite, including single-stranded DNA, hairpins, triplexes, R-loops, and i-motifs [107,108,109,110,111,112,113]. Topoisomerase II has been reported to recognize and cleave centromere alpha-satellite DNA in vitro [109], supporting the idea that these sequences can adopt complex, nonlinear structures that mimic the supercoiled substrate normally bound by this enzyme. Treatment with proteinase K causes loss of secondary loop structures [112], suggesting that the protein-mediated processes of replication and condensation may cause secondary structure formation. The predictions about the native secondary-structure-forming properties of the centromere [105,109] imply that DNA structures are an intrinsic feature of centromere repeats. Indeed, centromeres are binding sites for factors involved in managing DNA topology. The nuclease/helicase DNA2 was recently shown to preferentially bind to the centromere and promote centromeric replication [114]. Additionally, the hairpin-binding protein MSH2-6 is enriched at the centromere in Xenopus extract, although no increase in hairpins themselves was reported [115]. This same study identified an enrichment of many single-stranded DNA bubbles, another type of non-B DNA conformation and source of potential fragility on replicating centromere DNA [115]. It is likely that chromatinization by centromere-specific nucleosomes containing CENP-A as well as the enrichment of these proteins may be important to resolve complex secondary structures during centromeric replication and transcription, which may represent a particularly vulnerable time to DNA damage and instability.

These structures may be consequential in the maintenance of centromere structure and function and thus deserve further investigation. As in silico modeling improves, we may anticipate a more comprehensive analysis of predicted centromere structures; however, understanding the nature of the secondary DNA structures formed by centromeres in vivo remains of utmost importance. Further investigations will also be necessary to understand the many possible challenges secondary structures pose to structural and sequence integrity of the region.

### 4.3. Repeating the Repeats: The Challenges of Centromere Replication

The abundance of repetitive DNA and the aforementioned prevalence of secondary structures at centromeres create unique challenges during genome duplication that underscore the intrinsic fragility of centromeres in proliferating cells. In S-phase, slow-down or stalling of the active replisome may lead to replication fork collapse and a gap resulting in a DSB. Mass spectrometry analysis of Xenopus extracts reveals that centromeric DNA is enriched with a number of DNA damage factors [112]. Amongst the factors identified, the endonuclease MUS81 may be of particular interest. It is involved in fork progression during times of replicative stress and in the resolution of potentially mutagenic mitotic interlinks and recombination intermediates [116]. The RecQ-like helicase Werner (WRN) has also been observed to preferentially interact with centromeres and pericentromeres [117]. This association may be particularly emblematic of the challenges of centromeric DNA replication, as WRN plays a key role at stalled replication forks and at HR-generated Holliday junctions during DNA repair [118]. Because of the risks associated with DSB repair by recombination (Figure 1), limiting DNA damage at the repeats during S-phase is paramount for the maintenance of centromere stability. 

Given the challenges of replicating through repetitive sequences, centromeres may have developed specialized mechanisms that operate during S-phase. It is tempting to envisage a situation where CENP-A and its associated proteins facilitate efficient DNA replication at alpha-satellites in a similar manner to the telomere-binding proteins that are necessary for the progression of replication forks at telomeric repeats [119]. This may represent a novel role for centromere proteins additional to their mitotic functions toward kinetochore formation and chromosome segregation. Indeed, several CCAN components are constitutively associated with centromere loci throughout the cell cycle, and depletion in CENP-A, CENP-C, and CENP-T/W caused recombination and centromeres instability, while inducing chromosome missegregation did not [54]. Several scenarios by which CENP-A and associated proteins may maintain repeats stability during S-phase are possible: (1) Since naked alpha-satellite DNA is expected to form hairpin structures [109], depletion of CENP-A may lead to the formation of these secondary DNA structures during DNA replication, which would create obstacles for replication fork progression. Such stalled replication forks would lead to increased HR and promote centromere instability. (2) Alternatively, the presence of CENP-A on chromatin may directly facilitate the repair of lesions at centromeric repeats. Indeed, CENP-A is recruited to sites of DNA damage and seems to serve a function during DNA repair [120]. (3) A third hypothesis sees CENP-A promoting the recruitment of fork stabilizers and/or DNA repair factors, although no evidence of this was found in a recent mass-spectrometry analysis of CENP-A-bound factors in S-phase cells [49]. (4) CENP-A may regulate transcription, or transcription products stability, to avoid replication–transcription collision, RNA–DNA loops, or other sources of transcriptionally induced instability (discussed further in the next section). Whichever the mechanism, gaining a deeper understanding of centromere replication dynamics and challenges is a promising future area of study with relevance to the overall maintenance of genome stability in human cycling cells (Figure 2).

### 4.4. Breaking the Silence: Active Transcription of Centromere Alpha-Satellite Challenges Repeats Stability

Given that much of the centromere and pericentromere is largely devoid of protein-coding genes, the region was historically assumed to be transcriptionally silenced. Within the last decade, emerging evidence has shown that centromeres are not only actively transcribed, but that this transcription plays fundamental roles in the function and maintenance of centromere stability [28,121].

Transcription may represent a source of genomic instability through collision between replication and transcription forks, formation of secondary structures, or creation of cytotoxic DNA–RNA hybrids called R-loops [122], reviewed in [123] (Figure 1, left panel). The mutagenic potential of these R-loops is supported by the observation that mutants of fission yeast with excessive nuclear mRNA retention show increased genome instability [124]. Furthermore, loss of RNase H, the endonuclease which resolves R-loops, exacerbates this effect [124]. Paradoxically, DNA–RNA hybridization has also been shown to play a role in DNA repair (reviewed in [125]), and in vitro experiments show that this hybridization may facilitate DSB repair by bridging the broken DNA fragments in a Rad52-dependent manner during recombination [126].

Centromeres have been suggested [127,128], but not proven, to behave like fragile sites of the human genome. Common fragile sites are described as genomic loci where ongoing replication collides with the transcription machinery [129], which raises the tantalizing hypothesis that centromeres, also actively transcribed [121], have a similar behavior during S-phase. Similar to the aforementioned challenges during replication, transcription of centromeres during ongoing replication may be generated by the encounter of the two machineries within the same DNA molecule. DSBs may ensue from these conflicts between converging DNA-based transactions and lead to centromere instability (Figure 1).

Therefore, tight regulation of centromeric transcription is essential for the maintenance of genome stability. Transcription is important in the successful localization and stable incorporation of CENP-A [51,130] and in turn, CENP-A is needed for the maintenance of centromere stability [54]. The connection between centromeric transcription and CENP-A incorporation is further supported by the concurrent timing of RNA Pol II activity on the centromere and CENP-A localization in *Drosophila* [51]. Furthermore, active transcription appears to be important for the full integration of CENP-A onto the centromere, as CENP-A was more readily detached from untranscribed centromeres than from those that were robustly transcribed [51]. It is unclear if this effect is due to the transcript produced from RNA Pol II or from chromatin remodeling by RNA Pol II during transcription. Excess transcription, however, may also induce instability, as it can cause CENP-A to be removed [28,131]. While transcription’s effects on CENP-A retention have been assessed, the potential role of CENP-A, if any, in modulating transcription remains unclear.

At pericentromeres, heterochromatin and other epigenetic marks may also interact with transcription products to reinforce heterochromatin formation and subsequently regulate transcription in a positive feedback loop. For example, the transcriptionally repressive H3K9me3 mark may downregulate transcription, and the transcript produced from this region may interact with SUV39, the methyltransferase responsible for establishing and maintaining H3K9me3 [132]. Hypomethylation of satellite II repeats has been correlated with increased transcription from these regions, although hypomethylation alone is not sufficient to induce expression of these repeats [83]. Whether a similar interplay between epigenetic markers and transcriptional modulation happens at centromeres as at pericentromeres, and whether it may impact DNA stability, is unknown. Moreover, the control, timing, and magnitude of centromeric transcription should be further investigated to provide evidence for our aforementioned hypotheses regarding this novel source of fragility at centromeres.

### 4.5. Mitosis: A Tense Time for Centromeres

DNA damage response proteins, including 53BP1, RNF8, and RNF168, localize at centromeres during unperturbed mitoses [133,134,135], which suggests that the centromere may be poised for repair [133], potentially due to increased DNA damage during mitosis. In addition to stochastic damage, we can imagine two mitotic scenarios which may specifically challenge the integrity of centromeric DNA: (1) physical strain exerted on the alpha-satellite DNA by the spindle microtubules during the separation of the two sister chromatids in anaphase [134,136] or (2) the delayed resolution of concatenated DNA by Topo-II [137,138], which may be due to ongoing replication of centromere DNA into mitosis [48,139,140]. Indeed, like other genomic regions poor in protein-coding genes, centromeres replicate late in S-phase [48,139,140] and delays in replication (see Section 4.3: “Repeating the Repeats: The Challenges of Centromere Replication”) may lead to repeats being only partially duplicated in G2/M. Ultrafine bridges (UFBs), thin stretches of DNA connecting separating sisters, have been reported at centromeres in unperturbed cells [137,138], but it is unclear whether they represent physiological, rather than pathological, structures [141]. Furthermore, the current estimate of spindle force seems insufficient, perhaps by design, to sheer DNA [142]. After mitotic entry, however, breaks per se do not hinder mitotic progression [134] unless they interfere with the kinetochore, a complex multiprotein structure that mediates bioriented chromosome attachment to microtubules and/or satisfaction of the spindle assembly checkpoint (SAC) for faithful sister chromatids separation at each round of cell division (reviewed in [143]). Failure of one or more of these events or in the function of kinetochore proteins themselves can result in aberrant chromosome segregation [136,144]. Whether DNA damage and alterations in the alpha-satellite DNA sequence do affect kinetochore formation and function, beyond disruption of the docking site for the CENP-A nucleosome, is unclear. CENP-A nucleosomes have been extensively studied as the physical foundation for kinetochore assembly in mitosis [145], but future work may be able to tease apart a potential contribution of centromere DNA from the essential role of CENP-A and associated proteins in this process.

A clear correlation has emerged between numerical chromosome instability and an increase in centromeric damage, quantified by the presence of the phosphorylated version of the H2AX variant, γH2AX, a marker for DNA damage [136]. Missegregated chromosomes are, in turn, more susceptible to damage [146]. Our findings, however, indicate that induction of chromosome missegregation does not increase alpha-satellite instability, measured by the presence of C-SCE [54]. The discrepancy between Giunta and Funabiki (2017) [54] and Guerrero et al. (2010) [136] may be reconciled if the centromere damage repaired in the following G1 does not result in crossover between the sisters and, hence, is undetectable by the Cen-CO-FISH method [55]. Nonetheless, the connection between centromere damage, repeat alteration, and mitotic segregation remains an important outstanding question in the field.

### 4.6. Transposable Elements (TEs) at the Centromere: Friends or Foes?

TEs are DNA sequences that are capable of “jumping” around the genome. TEs contribute to both genomic instability, as their integration may result in the disruption of a functional DNA sequence, and genomic innovations, where novel sequence insertion may lead to novel function (reviewed in [16]). The role of TEs at the centromere is particularly interesting in the context of this review, given their appreciated contributions to genomic instability. Indeed, centromeres appear to be a site of preferential TE insertion [147], reviewed in [148]. The mechanism by which these TEs are recruited to the centromere is unclear, but it has been hypothesized that they may associate with CENP-A [149], reviewed in [16].

Most organisms have evolved complex regulatory mechanisms to silence transcription of TEs via DNA methylation and chromatin remodeling [150]. Although TEs transcription at centromeres may be silenced, it is not difficult to imagine that they may induce structural changes to the surrounding genomic region, at which the centromere may be particularly vulnerable due to a complex underlying secondary structure.

Most, if not all, transposons in the human genome have been inactive for the last 500 million years [151]; however, some retrotransposable (RT) elements, including the very prevalent LINE and SINE repeats which dominate the pericentromere of most human chromosomes, are still active [152]. Additional centromeric elements may also have their origins in RT elements, such as the CENP-B box [153] and the CENP-B protein [154]. These RT elements have been identified along much of the centromeric region, including within the core HOR sequences and between epialleles [147,148]. The role of these RT elements in the evolutionary history of the centromere is complex, but one which may serve to resolve some of the paradoxes about the evolutionary history of the centromere (reviewed in [16]).

In the context of our review, which highlights the prominent vulnerability of alpha-satellite DNA, it is perhaps unsurprising that these genomic elements which promote instability may be preferentially found at a locus which seems predisposed to high rate of sequence fragility. RT elements, however, have a role in promoting their own transcription, which may in turn promote centromeric stability (see Section 4.4: “Breaking the Silence: Active Transcription of Centromere Alpha-Satellite Challenges Repeats Stability”). Functional centromeres may form on artificial chromosomes which lack TE sequences, although they cannot form without active transcription [28]. It is therefore unclear whether transposons have a destabilizing role at the centromere or whether they rather contribute to centromere stability via transcription induction or other processes. Determining the relative balance between stabilization and destabilization effects of transposable and other elements at the centromere and the physiological and pathological consequences of these jumping DNA on alpha-satellite sequence and function remains an exciting area of research.

## 5. Future Directions: Novel Fragility of the Human Genome Specific to Centromeres

Centromere tandem DNA repeats represent a challenge for the maintenance of genomic stability, especially in cycling cells during normal cellular functions such as replication and transcription. However, mechanisms should be in place to alleviate these hurdles and maintain these important genomic loci stable over multiple generations. CENP-A depletion causes rearrangements of alpha-satellite DNAs [54], suggesting that CENP-A may be part of a wider centromere stability network of factors that works, directly or indirectly, to prevent repeats instability. Although CENP-A depletion causes chromosome missegregation and formation of micronuclei, chromosome missegregation and micronuclei formation by depleting Hec1 or through recovering from monastrol-induced monopolar spindle formation did not enhance rearrangements at centromeres [54]. Consistent with the role of CENP-A in directly preserving the integrity of alpha-satellite DNA repeats, reduction of the repeat has been observed at the spontaneously inactivated centromere in the chromosome that gained a neocentromere [155,156]. CENP-A and its associated proteins may contribute to centromere stability in several ways: (1) Efficient DNA replication at alpha-satellites via an analogous mechanism to telomere-binding proteins for efficient progression of replication fork at the terminal repeats [119]. (2) Since naked alpha-satellite DNA is expected to form hairpin structures [109], depletion of CENP-A may stabilize these secondary DNA structures, making it difficult for the replication fork to progress through the region. (3) Presence of CENP-A on chromatin may directly facilitate the repair of lesions at centromeric repeats, a hypothesis supported by the observation that CENP-A is recruited to sites of DNA damage and functions in DNA repair [120]. (4) Centromere compaction via CENP-A may contribute to suppression of rearrangements at the centromere, similar to shelterin-mediated telomere compaction implicated in suppression of DNA damage responses. Indeed, 3D super-resolution analyses of centromeres by Giunta and Funabiki (2017) suggest that CENP-A depletion causes an apparent increase in the chromatin volume occupied with alpha-satellites [54]. (5) CENP-A may modulate centromere transcription to avoid replication collision or reduce the likelihood of R-loop formation in an effort to limit centromere damage during S-phase and beyond (Figure 2). While many of the aforementioned hypotheses are inspired by the large body of work regarding telomeres, we cannot help but notice that the equally essential repetitive regions of human centromeres have lacked the same intense investigation from which their cousins at chromosome end have benefited. It is time to share the spotlight and shine it onto centromere repeats, with much still to learn.

## Figures and Tables

**Figure 1 genes-09-00615-f001:**
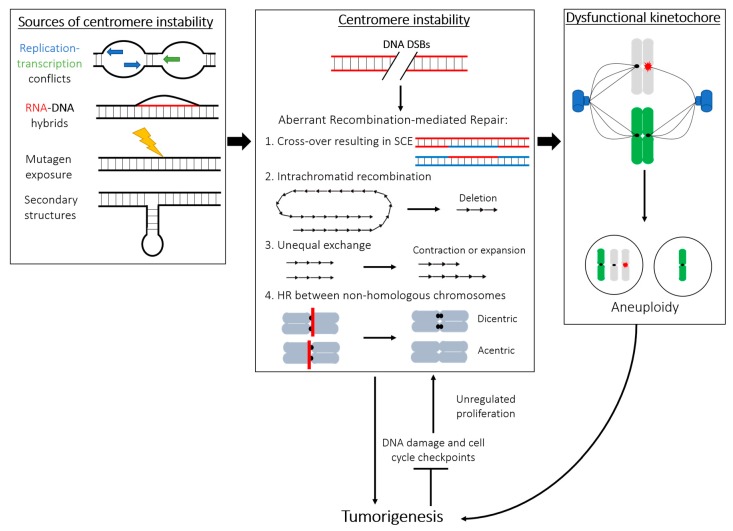
Centromere DNA instability as a contributor to aneuploidy and carcinogenesis. Left panel. Sources of centromere instability, such as replication–transcription conflicts, RNA–DNA hybrids forming R-loops, mutagen exposure and stochastic damage, as well as various secondary structures may underlie the inherent fragility of centromeres and propensity for DNA damage. Middle panel. These burdens may result in DNA double-stranded breaks (DSBs) formed at the centromere. Repair of these DSBs through recombination-dependent pathways, such as homologous recombination (HR), may disrupt centromere integrity in several ways: (1) Crossover between sister chromatids will lead to sister chromatid exchange (SCE), which has been reported at human centromeres. (2) Search for the homologous sequence may erroneously identify an identical or nearly identical sequence within the same chromatid downstream or upstream of the break site. Recombination between these two regions would result in DNA excision into an extrachromosomal circle and subsequent loss of alpha-satellite sequences. (3) Recombination may lead to unequal exchange, thereby introducing instability in the total size of the centromeric array. (4) HR at identical centromere sequences between different chromosomes can lead to the formation of dicentric and acentric chromosomes. Right panel. Rearrangements, changes, and especially loss, of alpha-satellite DNA may impact the centromere and, in turn, affect the formation of a functional kinetochore during mitosis. Centromere/kinetochore dysfunction increases the risk of chromosome missegregation and formation of aneuploid daughter cells. Aneuploidy can, in turn, promote tumorigenesis. Other routes may exist by which centromere instability promotes malignant transformation independent of kinetochore establishment, including altered transcription, delayed decatenation, structural integrity, etc. (see text in Section 4). Canonical DNA damage and cell division checkpoints become activated at specific stages during this process to prevent transformation. Inhibition of these failsafe pathways that restrict unregulated proliferative growth and continued cycling with subsequent increase in centromere/genome instability causes a short circuit with deleterious consequences that further promote mutagenesis and cancer formation.

**Figure 2 genes-09-00615-f002:**
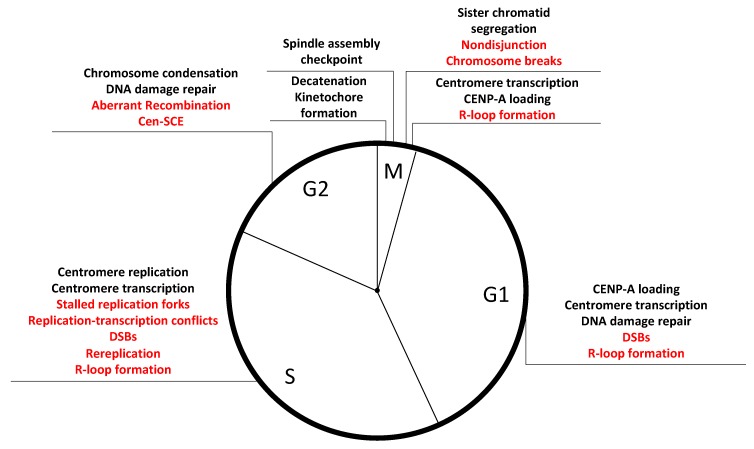
Cell-cycle-associated events that may destabilize human centromeres. Major cell cycle events and checkpoints are indicated at their respective location in the cell cycle in black, and potential sources of centromeric instability are shown in red. (G1) In the early stages of G1, centromeres are transcribed, and CENP-A is loaded. Close regulation of transcription is essential for loading CENP-A at the correct locus and in the correct amount, and incorrect loading of CENP-A may compromise centromere stability. (S) Replication of centromeric DNA may induce DNA instability due to stalled replication forks caused by unresolved secondary structures, re-replication of highly repetitive DNAs, and DSBs induced by replication fork collapse, torsional stress of secondary structure resolution, and/or unwinding. Centromere transcription in S-phase can lead to replication and transcription conflicts and collision. Transcripts may be an additional source of instability through the formation of mutagenic R-loops. (G2) As in S-phase, availability of a homologous sequence allows repair of DSBs by HR. Centromere DNA damage may be dealt with through aberrant repair pathways, either due to uneven or non-homologous recombination, leading to centromeric sister chromatid exchanges (C-SCE) between daughter DNA strands, sequence, and/or copy number alternation in the alpha-satellite array. (M) Before chromosome segregation, chromatids decatenation and kinetochore formation occurs, and every chromosome must satisfy the spindle assembly checkpoint. Nondisjunction at the centromere may lead to DNA breaks and aneuploidy of the daughter cells. Chromosomal breaks and arm loss may occur due to the physical stress of separating sister chromatids exerted on centromeres. Centromeric transcription and CENP-A loading occur in late telophase, the final stage of mitosis, which may lead to transcriptionally induced instability and R-loop formation.

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
