# Peer review of "Repetitive Fragile Sites: Centromere Satellite DNA as a Source of Genome Instability in Human Diseases"

_genes, 2018, doi:10.3390/genes9120615_

Round 1

Reviewer 1 Report

The manuscript by Black and Giunta is a review dealing with instability of centromeric DNA. After an introduction covering general considerations about centromeres, authors have chosen to focus on the consequences of centromere instability on human health and diseases, and reviewed all the potential causes for this instability.

The manuscript is very well written and its construction is very clear. It presents many very interesting ideas and an original point of view, which makes it suitable for publication in Genes. Nevertheless the manuscript may still be improved by correcting a few confusing sentences (see specific comments below) and by taking into account the following comments:

-it is is true that the presence of repeats has "precluded the complete linear assembly and annotation of all centromeres, thus far". Nevertheless, some important progresses have been made recently in the field, which pave the road to more quantitative functional analysis in the future and therefore should be mentionned: developpement of sequence models, assembly of the centromere of human chromosome Y...

-as most of the exemples concern different satellite DNAs from the human genome, it would be helpful to describe a bit better the organization of the different types of satellite sequences at human centromeres and pericentromeres (alpha satellite, satellite II, etc). 

-it is not completely true that "Within the pericentromere, alpha-satellite DNA is monomeric" (l.66-67), as in the case of epialleles, stretches of HOR are obviously in the pericentromeric regions.

-the part regarding cenpB and the resolution of the centromere paradox is described in two different places (l.73-78 and l.313-319). One would be largely enough.

-l.133-134: authors state that breaks occur more frequently at the centromere and pericentromeric regions than at other regions of the genome, citing their own work (a web protocol) and unpublished data. This idea is of great importance for this review and should be better introduced. Is the protocol citation an error? How did the authors estimate the frequencies?

-to illustrate the link between centromeres and cancer, authors insist on the instability of the repeats and on the quantity of cenpA. They mention the influence of dysregulated transcription on centromere stability but not the influence of centromere reorganization on gene transcription (see for example Fournier et al, EMBO Mol Med, 2, 159-171). I think this aspect is missing.

-bibliography is abundant, with a good mix of ancient (but important) references and very recent papers. Maybe the best references were not chosen for a few ideas (for example ref .6 is not the best to illustrate considerations about centromere size, and at l.386 ref. 17 refers to a very artificial system and may not be the best one). I would recommand the authors check the best matching references again.

-two figures are proposed. While the first one is just a remake of known ideas (a very similar figure can be found in ref 24 for example), the second one is new and informative. Maybe replacing figure 1 with another one highlighting the different molecular processes around centromeres would be more helpful (formation of secondary structures, transcription, etc).

In conclusion, I recommand the publication of this review in Genes after minor revisions have addressed some of these issues.

specific comments:

l.40: "observation". It is not very clear if this refers to the "presence of neocentromeres..." or to the "idea that this repetitive feature...". 

l.45-46: ommitted 10% of human DNA repeats: this formulation is confusing: is it 10% of the repeats or of the genome?

l.64: increasingly -> progressively?

l65: not sure if "disorganized" is the best word.

l.96: the pericentromere...is a site of constitutive heterochromatin: if the notion of constitutive heterochromatin is helpful, it must be explained.

l.261: not sure it is adapted to write about a "giant of centromere research" in this review (even if I share the idea with the author)

l.310: non-B-DNA forms ...   are rather prevalent in the human genome. Are they frequent enough to justify this formulation?

Author Response

The authors would like to thank both reviewers for raising several important points of reflection and giving us an opportunity to improve upon our manuscript based on their helpful comments. We have also improved upon the English redaction in this new version of the manuscript, as per the reviewers’ suggestion.

Reviewer #1

- It is true that the presence of repeats has "precluded the complete linear assembly and annotation of all centromeres, thus far". Nevertheless, some important progresses have been made recently in the field, which pave the road to more quantitative functional analysis in the future and therefore should be mentioned: development of sequence models, assembly of the centromere of human chromosome Y...

We completely agree with this point, which has been an oversight on our part. We have now corrected this by including reference to recent advancements in the field. See lines 51-54:

 “A linear assembly for the Y centromere has been achieved recently [1] and reference models for the other centromeres have been established [4,5, Miga et al., unpublished data], which may pave the way for future analysis of human centromere genetics.”

- As most of the examples concern different satellite DNAs from the human genome, it would be helpful to describe a bit better the organization of the different types of satellite sequences at human centromeres and pericentromeres (alpha satellite, satellite II, etc).

We have now added mention of the satellite II and III repeats at the human pericentromere and drawn a comparison between these repeats and alpha satellite DNA in lines 75-77:

“Additionally, other types of satellite repeats, such as the 5 bp satellite II and satellite III repeats, are abundant in the pericentromere of certain chromosomes [17, reviewed in 18]. In contrast, alpha-satellite DNA are present on every centromere [reviewed in 19].”

Due to length constrain, we have limited our edits to the aforementioned lines and pointed the readers to more comprehensive recent review on satellite DNAs, but we welcome feedback from the reviewer should further explanation still be required.

- It is not completely true that "Within the pericentromere, alpha-satellite DNA is monomeric" (l.66-67), as in the case of epialleles, stretches of HOR are obviously in the pericentromeric regions.

We have added the word predominantly in order to make the statement more general and to include examples of epialleles in the pericentromere.

“Within the pericentromere, alpha-satellite DNA is predominantly monomeric (no longer arranged into HORs) and interspersed with other sequences and repetitive elements, including the LINE and SINE retrotransposons”

We have also better defined the active centromere in our updated text at lines 60-62:

“The core centromere, made of highly homogenous HORs, is the functional site to enable chromosome division and spans between 0.5 to 5 Mb in size depending on the chromosome.”

- The part regarding cenpB and the resolution of the centromere paradox is described in two different places (l.73-78 and l.313-319). One would be largely enough.

We completely agree and have now erased the first description that was originally found at lines 73-78.

- l.133-134: authors state that breaks occur more frequently at the centromere and pericentromeric regions than at other regions of the genome, citing their own work (a web protocol) and unpublished data. This idea is of great importance for this review and should be better introduced. Is the protocol citation an error? How did the authors estimate the frequencies?

We thank the reviewer for pointing out this important error in citation. We now placed the correct reference here, Knutsen et al 2005 that shows over 50% of breaks and subsequent rearrangements happen at centromere in specific cancer lines. Our recent analyses using gamma-H2AX also confirmed this intrinsic fragility (unpublished data).

- To illustrate the link between centromeres and cancer, authors insist on the instability of the repeats and on the quantity of cenpA. They mention the influence of dysregulated transcription on centromere stability but not the influence of centromere reorganization on gene transcription (see for example Fournier et al, EMBO Mol Med, 2, 159-171). I think this aspect is missing.

We thank the reviewer, and have integrated this point in our review. We have added this reference and discussion to the role of centromere rearrangements on transcription. Given the high frequency of centromere rearrangements and transcriptional dysregulation in cancer, we think this is a valuable addition to our discussion of centromere instability in cancer. See lines 226-233:

“While the frequency of centromere rearrangements in cancer is well-appreciated, the mechanism of their malignancy is poorly understood. One possible explanation may lie in the perturbation of the epigenetic landscape of specific genomic regions triggering their subsequent transcriptional disregulation (Fouriner et al., 2016). Rearrangements within the pericentromeric DNA seem to disrupt not only the local heterochromatin but also contribute to long-range changes to gene expression (Fouriner et al., 2016), suggesting that centromere DNA instability may similarly induce changes in the chromatin state of the region and result in aberrant rates of transcription contributing to malignant transformation.”

- Bibliography is abundant, with a good mix of ancient (but important) references and very recent papers. Maybe the best references were not chosen for a few ideas (for example ref .6 is not the best to illustrate considerations about centromere size, and at l.386 ref. 17 refers to a very artificial system and may not be the best one). I would recommend the authors check the best matching references again.

We have gone through thoroughly revising our references. Altemose, et al., 2014, has now been referenced alongside Fowler et al., 1987 and van Dekken et al., 1990, more original works showing the significant variation in satellite array size between individuals. We welcome any additional feedback from the reviewer to further improve our citations.

- Two figures are proposed. While the first one is just a remake of known ideas (a very similar figure can be found in ref 24 for example), the second one is new and informative. Maybe replacing figure 1 with another one highlighting the different molecular processes around centromeres would be more helpful (formation of secondary structures, transcription, etc).

This is an excellent suggestion. We have worked hard to re-integrate the ideas from Figure 1 into a new, more comprehensive figure now proposed in the current draft.

Specific comments:

l.40: "observation". It is not very clear if this refers to the "presence of neocentromeres..." or to the "idea that this repetitive feature...".

We have reworded for clarity:

“However, the presence of neocentromeres in non-repetitive genomic loci strongly supports the idea that this repetitive feature may not be essential for maintaining centromere function [reviewed in 2]:”

l.45-46: ommitted 10% of human DNA repeats: this formulation is confusing: is it 10% of the repeats or of the genome?

We have clarified that The Human Genome Project has omitted over 10% of the genome, including large proportion of DNA repeats. This now reads (lines 45-46):

“Indeed, The Human Genome Project, marked as completed in 2003 [3], actually omitted over 10% of the human genome including large portions of the centromere and other repetitive elements.”

l.64: increasingly -> progressively?

We thank the reviewer for suggesting a better word here:

“In the region flanking the centromere, the highly homogeneous alpha-satellite becomes progressively more heterogeneous and unstructured.”

l65: not sure if "disorganized" is the best word.

We have now changed that to unstructured, as follows:

“In the region flanking the centromere, the highly homogeneous alpha-satellite becomes progressively heterogeneous and unstructured.”

l.96: the pericentromere...is a site of constitutive heterochromatin: if the notion of constitutive heterochromatin is helpful, it must be explained.

An explanation of constitutive heterochromatin has been added as per below:

“Constitutive heterochromatin describes the highly compacted and transcriptionally repressed chromatin conformation found in repetitive and gene-poor genomic regions, such as the telomere and pericentromere (reviewed in Saksouk et al., 2015).”

l.261: not sure it is adapted to write about a "giant of centromere research" in this review (even if I share the idea with the author)

Phrase has been deleted, we agree that the well-deserved tribute was out of place.

l.310: non-B-DNA forms ...   are rather prevalent in the human genome. Are they frequent enough to justify this formulation?

We have modified this sentence for clarity and have included reference to Kasinathan & Henikoff 2018. Additionally, we have added reference to a comprehensive review on the subject (Zhao et al., 2010).

Reviewer 2 Report

The article titled "Uncommon Fragile Sites: Centromere repetitive DNA as a source of genome instability in human diseases" by Black and Giunta is a comprehensive review that makes an interesting read.  It is an appropriate combination of discussion on basic centromere biology and medical relevance.  The general organization is sound and easy to follow.  Overall, it provides a valuable summary of the centromere research field.  As such, I have very limited concerns detailed as follows.

The usage of "uncommon fragile sites" has very specific connotations with respect to the chromosome fragile site field and may confuse the reader.  The word "uncommon" presumably sets it apart from "common" fragile sites.  However, "common fragile sites" are contrasted by "rare fragile sites" in a sense that the former is an intrinsic feature of DNA sequences shared by all individuals, while the latter refers to mutations in specific human populations.  Centromere fragility is not rare, based on the definition given.  In fact, it should be categorized as common fragile sites, although the authors seem to contend that on page 11, line 413: "Centromeres have been suggested, but not proven, to behave like fragile sites of the human genome".  No reference was given here and it would be important for the authors to explain their argument in more details because it is such a central theme of the review.  Otherwise, the authors are better advised to select a more precise adjective than "uncommon", or, at a minimum, put quotation marks around it.

Figure 1 serves little purpose in assisting the reader understanding key concepts in the review. HR, in this reviewer's opinion, should be quite ingrained in an average reader's mind that it may not require such a detailed outline.  Instead, the same space could be more effectively utilized to illustrate other concepts in the review.  For instance, DNA secondary structure formation, in all its varieties, as impediments to replication and as regulators for transcription can be illustrated here.  Alternatively, the histone modifications in the context of CENP-A binding would be easier to follow with some visuals.

In section 4.1, the general concept is HR at centromere appears to be inhibited.  However, the authors put forth "...it should not be surprising that centromeres may suffer DNA damage, including DSBs, which should be repaired by HR."  DSB repair in eukaryotes is more often repaired by non-homologous end joining than HR.  Is centromere an exception?  More in-depth discussion on this subject is needed here.

Discussion on cohesins and kinetochore proteins, which are central proteins that interact with centromeres, appear to be left out of the discussion here.  Perhaps a brief section on these subjects would make it even more comprehensive.

Overall, the style of writing at times feels not straightforward. For instance,

page 3, line 114: "The reestablishment of CENP-A during telophase/early G1...".  What do the authors mean by "reestablishment"?  Are they referring to the transition of de novo CENP-A to fully chromatin-incorporated CENP-A?  If so, it would serve the reader better by stating that instead.

page 5, line 182: "...is the reduction or delocalization of CENP-A...".  What does delocalization mean? Does it mean distribution?

Some grammatical errors are listed below:

Abstract: "...and complete annotation of all human centromeres is still incomplete".  "complete" and "incomplete" are awkward in the same sentence.  Perhaps change it to "...and complete annotation of all human centromeres still falls short".

Page 4, line 150: "ICF syndrome resulting from mutation of these three genes is dissimilar to the DNA methylation profile found in DNMT3b-mutated ICF patients..." should probably read "DNA methylation profile in ICF patients with mutations in any of these three genes is different from that in ICF patients with DNMT3b mutation."

Author Response

The authors would like to thank both reviewers for raising several important points of reflection and giving us an opportunity to improve upon our manuscript based on their helpful comments. We have also improved upon the English redaction in this new version of the manuscript, as per the reviewers’ suggestion.

Reviewer #2

- The usage of "uncommon fragile sites" has very specific connotations with respect to the chromosome fragile site field and may confuse the reader.  The word "uncommon" presumably sets it apart from "common" fragile sites.  However, "common fragile sites" are contrasted by "rare fragile sites" in a sense that the former is an intrinsic feature of DNA sequences shared by all individuals, while the latter refers to mutations in specific human populations.  Centromere fragility is not rare, based on the definition given.  In fact, it should be categorized as common fragile sites, although the authors seem to contend that on page 11, line 413: "Centromeres have been suggested, but not proven, to behave like fragile sites of the human genome".  No reference was given here and it would be important for the authors to explain their argument in more details because it is such a central theme of the review.  Otherwise, the authors are better advised to select a more precise adjective than "uncommon", or, at a minimum, put quotation marks around it.

We thank the reviewer for this point. We have renamed the title to better define them:

“Repetitive Fragile Sites: Centromere satellite DNA as a source of genome instability in human diseases”

The intrinsic fragility of centromeres is analogous to common fragile sites, as noted by reviewer#2 comment above, with additional features and potential sources of instability. We have also now better defined our stance on page 10 lines 410-413, saying:

“Centromeres have been suggested [127-128], but not proven, to behave like fragile sites of the human genome. Common fragile sites are described as genomic loci where ongoing replication collides with the transcription machinery [128], which raises the tantalizing hypothesis that centromeres, also actively transcribed [122], have a similar behavior during S-phase.”

- Figure 1 serves little purpose in assisting the reader understanding key concepts in the review. HR, in this reviewer's opinion, should be quite ingrained in an average reader's mind that it may not require such a detailed outline.  Instead, the same space could be more effectively utilized to illustrate other concepts in the review.  For instance, DNA secondary structure formation, in all its varieties, as impediments to replication and as regulators for transcription can be illustrated here.  Alternatively, the histone modifications in the context of CENP-A binding would be easier to follow with some visuals.

This is an excellent suggestion. We have worked hard to re-integrated the ideas from Figure 1 into a completely new, more comprehensive Figure 1.

- In section 4.1, the general concept is HR at centromere appears to be inhibited.  However, the authors put forth "...it should not be surprising that centromeres may suffer DNA damage, including DSBs, which should be repaired by HR."  DSB repair in eukaryotes is more often repaired by non-homologous end joining than HR.  Is centromere an exception?  More in-depth discussion on this subject is needed here.

The original sentence did not express clearly the concept of stochastic breaks, and that at least a proportion of them are repaired by HR during S and G2 phases. The new sentence now better explain this concept:

During each round of replication, unperturbed cells suffer over 40 DNA DSBs [102], of which at least half are repaired by HR in S-phase and G2, when the sister chromatid is available as template for repair [reviewed in 103]. Thus, it should not be surprising that centromeres stochastically suffer DNA damage, including DSBs, which may be repaired by HR.

Discussion on cohesins and kinetochore proteins, which are central proteins that interact with centromeres, appear to be left out of the discussion here.  Perhaps a brief section on these subjects would make it even more comprehensive.

We appreciate the reviewer comment. Exploring the topic of protein players, especially cohesins, in centromere DNA stability, is very important; however, due to space restrictions, we feel it may be beyond the immediate scope of this review.

- Overall, the style of writing at times feels not straightforward. For instance,

- page 3, line 114: "The reestablishment of CENP-A during telophase/early G1...".  What do the authors mean by "reestablishment"?  Are they referring to the transition of de novo CENP-A to fully chromatin-incorporated CENP-A?  If so, it would serve the reader better by stating that instead.

We thank the reviewer for this suggestion. We have clarified this statement by replacing the phrase “reestablishment of CENP-A” with “incorporation of newly synthesized CENP-A into centromeric chromatin”

- page 5, line 182: "...is the reduction or delocalization of CENP-A...".  What does delocalization mean? Does it mean distribution?

We have rewritten this statement to avoid confusion with the word delocalization. Instead, we have changed the word to mislocalization and clarified that mislocalized CENP-A refers to CENP-A that is present in non-centromeric regions. The sentence now reads:

“One possible explanation for the observed correlation between centromeric instability and cells undergoing senescence is the reduction or mislocalization of CENP-A away from the centromere, which has been reported in aging and cellular senescence”

- Some grammatical errors are listed below:

Abstract: "...and complete annotation of all human centromeres is still incomplete".  "complete" and "incomplete" are awkward in the same sentence.  Perhaps change it to "...and complete annotation of all human centromeres still falls short".

We have replaced the word incomplete with lacking. We thank the reviewer for the careful reading and opportunity to improve upon our manuscript and its clarity.

Page 4, line 150: "ICF syndrome resulting from mutation of these three genes is dissimilar to the DNA methylation profile found in DNMT3b-mutated ICF patients..." should probably read "DNA methylation profile in ICF patients with mutations in any of these three genes is different from that in ICF patients with DNMT3b mutation."

We have now incorporated the reviewer’s suggestion.
